# LRP1-Mediated AggLDL Endocytosis Promotes Cholesteryl Ester Accumulation and Impairs Insulin Response in HL-1 Cells

**DOI:** 10.3390/cells9010182

**Published:** 2020-01-10

**Authors:** Virginia Actis Dato, Aleyda Benitez-Amaro, David de Gonzalo-Calvo, Maximiliano Vazquez, Gustavo Bonacci, Vicenta Llorente-Cortés, Gustavo Alberto Chiabrando

**Affiliations:** 1Departamento de Bioquímica Clínica, Facultad de Ciencias Químicas, Universidad Nacional de Córdoba, Córdoba 5000, Argentina; vickyactisdato@gmail.com (V.A.D.); mvazquezgc@gmail.com (M.V.); gustavobonacci@gmail.com (G.B.); 2Consejo Nacional de Investigaciones Científicas y Técnicas (CONICET), Centro de Investigaciones en Bioquímica Clínica e Inmunología (CIBICI), Córdoba 5000, Argentina; 3Institute of Biomedical Research of Barcelona (IIBB)-Spanish National Research Council (CSIC), 08025 Barcelona, Spain; aleyda.benitez.amaro@gmail.com (A.B.-A.); david.degonzalo@gmail.com (D.d.G.-C.); 4Biomedical Research Institute Sant Pau (IIB Sant Pau), 08025 Barcelona, Spain; 5CIBERCV, Institute of Health Carlos III, 28029 Madrid, Spain

**Keywords:** glucose, signaling, lipid, membranes, intracellular traffic

## Abstract

The cardiovascular disease (CVD) frequently developed during metabolic syndrome and type-2 diabetes mellitus is associated with increased levels of aggregation-prone small LDL particles. Aggregated LDL (aggLDL) internalization is mediated by low-density lipoprotein receptor-related protein-1 (LRP1) promoting intracellular cholesteryl ester (CE) accumulation. Additionally, LRP1 plays a key function in the regulation of insulin receptor (IR) and glucose transporter type 4 (GLUT4) activities. Nevertheless, the link between LRP1, CE accumulation, and insulin response has not been previously studied in cardiomyocytes. We aimed to identify mechanisms through which aggLDL, by its interaction with LRP1, produce CE accumulation and affects the insulin-induced intracellular signaling and GLUT4 trafficking in HL-1 cells. We demonstrated that LRP1 mediates the endocytosis of aggLDL and promotes CE accumulation in these cells. Moreover, aggLDL reduced the molecular association between IR and LRP1 and impaired insulin-induced intracellular signaling activation. Finally, aggLDL affected GLUT4 translocation to the plasma membrane and the 2-NBDG uptake in insulin-stimulated cells. We conclude that LRP1 is a key regulator of the insulin response, which can be altered by CE accumulation through LRP1-mediated aggLDL endocytosis.

## 1. Introduction

Metabolic syndrome (MS) and type-2 diabetes mellitus (T2DM) are chronic metabolic disorders and prevalent diseases adversely impacting health worldwide, which are characterized by increased levels of circulating glucose, insulin, atherogenic lipoprotein subfractions, and inflammatory cytokines [1,2]. These mediators of metabolic disease modify multiple molecular pathways, such as ceramide-induced apoptosis, increased formation of reactive oxygen species, mitochondrial dysfunction, and endoplasmic reticulum stress, which drive multiple alterations in different cell types, including cardiomyocytes [1,3]. Cardiomyocytes are insulin-responsive cells and impaired insulin-signaling are associated with heart failure [4]. The excessive accumulation of neutral lipids is a phenomena associated with altered insulin signaling activation in hepatocytes, muscle cells, and myocardium [5,6]. However, the molecular mechanisms underlying the link between impaired insulin action and lipid accumulation are largely unknown.

Patients with MS and T2DM usually present increased levels of aggregation-prone small LDL particles in addition to other anomalies in the lipoprotein profile [7]. These small LDL particles have a high tendency to become retained and aggregated (aggLDL) in extracellular spaces, but a small proportion of aggLDL has also been detected in circulation [7,8,9]. Whereas native LDL are recognized by the LDL receptor (LDLR), aggLDL are taken up by the LDL receptor-related protein 1 (LRP1) [10]. Moreover, LRP1 is involved in intracellular CE accumulation in muscle cells [11,12,13], which promotes a significant up-regulation of the *lrp1* gene expression mediated by the sterol regulatory element-binding proteins (SREBP) down-regulation [14,15,16].

It is known that LRP1 regulates the intracellular traffic of insulin-responsive vesicles containing the glucose transporter GLUT4 (GSV for GLUT4 storage vesicles) in fat and muscle cells [17]. These vesicles are trafficked and fused with the plasma membrane (PM) under insulin stimulus, through a mechanism dependent on the activation of the PI_3_K (phosphatidylinositol-3-kinase)/Akt pathway [18,19]. LRP1 depletion in GSV substantially reduces GLUT4 sorting to the PM promoting decreased glucose uptake [20]. In addition, it has been shown in hepatic and neuron-specific LRP1 knock-out mice that this receptor interacts with insulin receptor (IR) and regulates its intracellular signaling in neurons and hepatocytes [21,22]. Recently, we found that the blockage of LRP1 exocytosis towards the PM affected the insulin-induced intracellular signaling in retinal Müller glial cells [23]. These data suggest that LRP1 plays a key role in the insulin response in different types of cells and tissues.

On the basis of these previous results, we hypothesize that LRP1 is involved both in CE accumulation and insulin response impairment in cardiomyocytes treated with aggLDL. Thus, the main objective of the present study was to evaluate the role of LRP1 in the aggLDL-mediated intracellular CE accumulation and in the impairment of insulin response evaluated through the insulin signaling activation, GLUT4 trafficking and glucose uptake in HL-1 cardiomyocytes-derived cell line. Herein, we demonstrated that LRP1 mediates the aggLDL binding and endocytosis, promoting CE accumulation in these cells. The aggLDL/LRP1 complex was accumulated in early endosomes [EEA1^+^] but not in other recycling vesicles such as TGN [TGN46^+^] or recycling endocytic compartments [Rab4^+^ and Rab11^+^]. Finally, aggLDL-treated HL-1 cells showed a decreased insulin response, which was evidenced by: (i) reduced molecular association between LRP1 and IR; (ii) decreased insulin-induced intracellular signaling (IR, Akt, and AS160 phosphorylation); (iii) impaired GLUT4 translocation to the PM; and (iv) reduced extracellular glucose uptake.

## 2. Material and Methods

### 2.1. HL-1 Cardiomyocyte-Derived Cell Line, Cultures, and Reagents

The murine HL-1 cardiomyocyte-derived cell line was generated by Dr. W.C. Claycomb (Louisiana State University Medical Centre, New Orleans, LA, USA) [24,25]. HL-1 cells were maintained in Claycomb Medium (Sigma-Aldrich, St. Louis, MO, USA) supplemented with 10% fetal bovine serum (FBS) (Invitrogen, Buenos Aires, Argentina), 100 µM nor-epinephrine (Sigma-Aldrich, St. Louis, MO, USA), 100 units/mL penicillin, 100 g/mL streptomycin (Invitrogen, Buenos Aires, Argentina), and L-glutamine 2 mM (GlutaMAX from Invitrogen, Buenos Aires, Argentina) in plastic dishes, coated with 12.5 g/mL fibronectin (Sigma-Aldrich, St. Louis, MO, USA) and 0.02% gelatin, in a 5% CO_2_ atmosphere at 37 °C. Insulin was from Apidra^®^ Solostar^®^ 100 U/mL (Sanofi-Aventis, Germany). Rabbit anti-IR (cs4B8), rabbit anti-pIR (Tyr1361, cs84B2), and rabbit anti-Akt (#9272) monoclonal antibodies were from Cell Signaling Technology (Beverly, MA, USA). Rabbit anti-pAkt (Ser473, #07-789) antibody was from Merck KGaA (Darmstadt, Germany). Rabbit anti-AS160 (#ab24469) and rabbit anti-pAS160 (Thr642, #ab65753) antibodies were from Abcam (Cambridge, MA, USA). Mouse monoclonal anti-β-actin (#A2228) was from Sigma-Aldrich (St. Louis, MO, USA). Mouse monoclonal anti-APT1A1 (M7-PB-E9) was from ThermoFisher Scientific (Rockford, IL, USA). Rabbit anti-LRP1 (ab92544), mouse monoclonal anti-LRP1 (#ab28320), rabbit anti-GLUT4 (#ab654), rabbit anti-EEA1 (#ab2900), rabbit anti-Rab4 (#ab13252), rabbit anti-Rab11 (#ab65200), and rabbit anti-TGN46 (#ab50595) monoclonal antibodies being purchased from Abcam (Cambridge, MA, USA). Immunofluorescences were performed with the secondary antibodies raised in goat anti-rabbit IgG conjugated with Alexa Fluor 647, 594 or 488, and anti-mouse IgG conjugated with Alexa Fluor 594 or 488 (diluted 1/800) (Invitrogen, Buenos Aires, Argentina). GST-RAP was expressed and purified as described elsewhere [26] and used without further modification. In this work, 400 nM GST-RAP was used to inhibit the binding of aggLDL to LRP1.

### 2.2. LDL Isolation and Modification

Total LDL (d 1.019–d 1.063 g/mL) was isolated by sequential ultracentrifugation using KBr gradients, in the density range of 1.019–1.063 g/mL, from six different pools of plasma of normocholesterolemic volunteers. For the protein quantification we used a Pierce kit (#23225, ThermoFisher Scientific (Rockford, IL, USA). Apolipoprotein B integrity was tested by SDS-PAGE in 10% acrylamide gels. Aggregated LDL was generated by vortexing LDL in PBS 1X for 5 min at room temperature [27], and then was centrifuged at 10,000 rpm for 10 min to precipitate the aggLDL. Finally, aggLDL was suspended in PBS 1X to a protein concentration of 1 mg/mL. The ultrastructure of aggLDL obtained by vortexing was similar to that of LDL modified by versican, one of the main chondroitin sulfate proteoglycans structuring the arterial intima [28].

### 2.3. DiI-Labeling of LDL

DiI (1,1-dioctadecyl-3,3,3,3-tetramethylindocarbocyanine, Invitrogen, Buenos Aires, Argentina) is a lipophilic long-chain dialkylcarbocyanine that binds to lipoproteins and emits fluorescence at 565 nm. The LDL (1 mg/mL) were incubated for 12 h with DiI in a proportion of 3 μL per 1 mg of lipoprotein in PBS 1X at 37 °C, exhaustively dialyzed in PBS 1X for 24 h to eliminate the excess of free DiI and filtered through a 0.22 μm filter. Finally, these stained LDL were aggregated mechanically by vortexing.

### 2.4. Lipid Extraction and Determination of CE, TG, and FC Content

HL-1 cells were serum-starved overnight and then exposed to aggLDL (100 μg/mL) for 8 h. Next, cells were exhaustively washed and harvested in NaOH 0.1 M. Lipids were extracted with dichloromethane/methanol [1:2] and cholesteryl esters (CE), free cholesterol (FC) and triglycerides (TG) content was analyzed by thin layer chromatography. The organic solvent was removed under N2 stream, the lipid extract was dissolved in dichloromethane and one aliquot was partitioned by thin layer chromatography (TLC) performed on silica G-24 plates. Different concentrations of standards (a mixture of cholesterol, cholesterol palmitate and triglycerides) were applied to each plate. The chromatographic solution was heptane/diethylether/acetic acid (74:21:4, *v*/*v*/*v*). The developing was carried out with a 5% solution of phosphomolybdic acid/5% sulfuric acid in ethanol and heating 7 min at 100 °C. The spots corresponding to CE, TG, and FC were quantified by densitometry against the standard curve of cholesterol palmitate, triglycerides, and cholesterol, respectively, using a computing densitometer.

### 2.5. Western Blot Analysis

HL-1 cells were cultured as above and cell protein extracts were prepared using RIPA lysis buffer (50 mM Tris-HCl pH 8.0, 150 mM NaCl, 1% Triton X-100, 0.5% sodium deoxycholate, 0.1% SDS, 1 mM PMSF, 10 mM sodium ortho-vanadate, and protease inhibitor cocktails (Sigma-Aldrich, St. Louis, MO, USA). Forty micrograms of cell protein extracts were diluted in sample buffer 5X with DTT (dithiothreitol) and then heated for 5 min at 95 °C. Electrophoresis on 10% SDS-polyacrylamide gels was applied and proteins were electrotransferred to a nitrocellulose membrane (GE Healthcare Life Science, Amsterdam, The Netherlands). Nonspecific binding was blocked with 5% non-fat dry milk in a Tris-HCl saline buffer containing 0.01% Tween 20 (TBS-T) for 60 min at room temperature. The membranes were incubated overnight at 4 °C with diluted primary antibodies and secondary antibodies raised in goat anti-mouse IgG IRDye^®^ 680CW and goat anti-rabbit IgG IRDye^®^ 800CW (LI-COR Biosciences, Lincoln, NE, USA) diluted 1/10,000 for 1 h at room temperature. The specific bands were revealed using Odyssey CLx near-infrared fluorescence imaging system (LI-COR Biosciences, Lincoln, NE, USA).

### 2.6. Confocal Microscopy

HL-1 cells were cultured as above on cover glass. After different stimulus the cells were washed with PBS 1X, fixed with 4% paraformaldehyde (PFA), quenched with 50 mM NH_4_Cl, permeabilized for 30 min with 0.5% (*v*/*v*) saponin, blocked with 2% bovine serum albumin (BSA) and incubated with the respective primary antibody (diluted from 1/100 to 1/250) for 1 h, and revealed with a secondary antibody conjugated with Alexa Fluor (1/800) and Hoechst 33,258 colorant (1/2000) for 1 h. Finally, the cells were mounted on glass slides with Mowiol 4–88 reagent from Calbiochem (Merck KGaA, Darmstadt, Germany). For co-localization analyses, fluorescent images were obtained with an Olympus FluoView FV1200 confocal laser scanning biological microscope (Olympus, NY, USA). Whole cells were scanned and optical sections were obtained in 0.25-μm steps perpendicular to the *z*-axis, with images being processed using the FV10-ASW Viewer 3.1 (Olympus, NY, USA) and quantified by ImageJ software (NIH, Bethesda, MD, USA).

### 2.7. Real Time-PCR

The cells were serum-starved overnight and then exposed to the different stimuli and treated with the TRIzol^®^ reagent (Invitrogen, Buenos Aires, Argentina) for total RNA extraction. One μg of total RNA was reverse transcribed in a total volume of 20 μL using random hexaprimers and reverse transcriptase. The PCR primers listed below were used to quantify the transcripts of LRP1, and β-actin. The results were normalized to RT-PCR products of β-actin transcripts. Transcripts were quantified by real-time qRT-PCR (ABI 7500 Sequence Detection System, Applied Biosystems, Foster City, CA, USA) using Sequence Detection software v1.4. Amplification conditions included a warm start at 95 °C for 10 min, followed by 40 cycles at 95 °C for 15 s and 60 °C for 1 min. Specificity was verified by fusion curve analysis and electrophoresis in 2% agarose gel with fluorescence detection with SYBR^®^ Safe DNA (Invitrogen, Buenos Aires, Argentina). Relative gene expression was calculated according to the 2-Ct method. Each sample was analyzed in triplicate. No amplification was observed in the PCRs using water or RNA samples incubated without reverse transcriptase during cDNA synthesis.

Sequences of mouse primers:

*lrp1* forward 5′-TGGAGCAGATGGCAATCGACT-3′

*lrp1* reverse 5′-CGAGTTGGTGGCGTAGAGATAGTT-3′

*β-actin* forward 5′-AAATCTGGCACCACACCTTC-3′

*β-actin* reverse 5′-GGGGTGTTGAAGGTCTCAAA-3′

### 2.8. Immunoprecipitation (IP) Assays

HL-1 cells were cultured as above, they were treated with or without aggLDL (100 μg/mL) for 8 h and then stimulated or not with 100 nM of insulin for 30 min. Cell protein extracts were prepared using RIPA lysis buffer and incubated for 2 h at 4 °C with rabbit anti-LRP1 monoclonal antibody or rabbit non-immune IgG as IP control (2 μg/200 μg of total proteins). Then, these were incubated overnight at 4 °C with protein A-conjugated agarose beads following the manufacturer’s procedure (sc-2001; Santa Cruz Biotechnology, Santa Cruz, CA, USA) and the proteins were separated and treated for Western blot using rabbit anti-IR (1/1000) and rabbit anti-LRP1 (1/10,000) monoclonal antibodies. Secondary antibodies raised in goat anti-mouse IgG IRDye^®^ 680CW and goat anti-rabbit IgG IRDye^®^ 800CW (LI-COR Biosciences, Lincoln, NE, USA) diluted 1/10,000 for 1 h at room temperature. The specific bands were revealed using Odyssey CLx near-infrared fluorescence imaging system (LI-COR) and were quantified by densitometric analysis using Image Studio Software (LI-COR Biosciences, Lincoln, NE, USA). As loading control of total protein extracts β-actin was used.

### 2.9. Biotin-Labeling Cell Surface Protein Assay

HL-1 cells were cultured as above and then treated with or without aggLDL (100 μg/mL) for 8 h, followed by the stimulation with 100 nM of insulin for 30 min. To determine the level of LRP1 and GLUT4 at the cell surface a biotin-labeling protein assay (EZ-Link™ Sulfo-NHS-SS-Biotin (cat: 21331), Thermo Scientific, Rockford, IL, USA) was used. Briefly, cells were incubated for 2 h at 4 °C with a 0.12 mg/mL of biotin solution. Then, were incubate with 0.1 mM glycine solution for 30 min and washed with PBS 1X three times to remove the excess of biotin, after which corresponding cell lysates were generated. The biotinylated proteins were pulled down using streptavidin-conjugated agarose beads (Pierce™ Streptavidin Agarose (cat: 20353), Thermo Scientific, Rockford, IL, USA) for 2 h at room temperature. The biotinylated-plasma membrane proteins and total proteins (corresponding to 10% of proteins incubated with agarose beads) were eluted by adding sample buffer 6X, DTT 1M and heating at 100 °C for 5 min, and then treated for Western blot using rabbit anti-LRP1 (1/10,000), rabbit anti-GLUT4 (1/2000), mouse monoclonal anti-ATP1A1 (1/1500), or mouse monoclonal anti-β-actin (1/5000) monoclonal antibodies. The membranes were incubated overnight at 4 °C with diluted primary antibodies and secondary antibodies raised in goat anti-mouse IgG IRDye^®^ 680CW and goat anti-rabbit IgG IRDye^®^ 800CW (LI-COR Biosciences, Lincoln, NE, USA) diluted 1/10,000 for 1 h at room temperature. The specific bands were revealed using Odyssey CLx near-infrared fluorescence imaging system (LI-COR Biosciences, Lincoln, NE, USA) and were quantified by densitometric analysis using Image Studio Software (LI-COR Biosciences, Lincoln, NE, USA). As loading control of PM protein and total protein extracts biotinylated-ATP1A1 and β-actin were used, respectively. The content of each biotinylated protein in the PM was analyzed by densitometry and related to biotinylated-ATP1A1 protein.

### 2.10. 2-NBDG Uptake Assay

HL-1 cells were cultured as above, and they were treated with or without aggLDL (100 μg/mL) for 8 h and then stimulated or not with 100 nM of insulin for 30 min together with 80 μΜ of 2-Deoxy-2-[(7-nitro-2,1,3-benzoxadiazol-4-yl) amino]-D-glucose (2-NBDG solution; Sigma-Aldrich, St. Louis, MO, USA). The 2-NBDG is a glucose fluorescent analog that is not metabolized and displays emission maxima of 540 nm. After stimulus, cells were washed with PBS 1X, fixed with 4% of paraformaldehyde, quenched with 50 mM NH_4_Cl, permeabilized for 30 min with 0.5% (*v*/*v*) saponin, blocked with 2% BSA and incubated with Hoechst 33,258 colorant (1/2000) for 1 h. Finally, cells were mounted on glass slides with Mowiol 4–88 reagent. Fluorescent images were obtained with an Olympus FluoView FV1200 confocal laser scanning biological microscope (Olympus, NY, USA). Whole cells were scanned and optical sections were obtained in 0.25-μm steps perpendicular to the *z*-axis, with images being processed using the FV10-ASW Viewer 3.1 (Olympus, NY, USA) and the total fluorescence in the whole cell area was quantified by ImageJ software. For microscope quantification of the co-localization level, a JACoP plug-in from ImageJ software (NIH, Bethesda, MD, USA) was used.

### 2.11. Statistical Treatment of Data

At least 50 cells/condition were analyzed and the averages of the vesicle percentages containing both proteins were calculated using the Manders’ coefficients and compared by the Student’s t-test. For Western blot and cell-surface protein detection assay, the data was expressed as the mean ± SEM and comparisons between two groups were performed using a paired t-test. Results from more than two groups were analyzed by one-way ANOVA followed by Dunnett’s post-hoc analysis (GraphPad Prism 5.0, San Diego, CA, USA). Values of *p* < 0.05 were considered to be significant.

## 3. Results

### 3.1. LRP1 Mediates Intracellular CE Accumulation by AggLDL Endocytosis

It has been reported that LRP1 is the cell surface receptor responsible for the binding and endocytosis of aggLDL and CE accumulation in VSMCs, macrophages and HL-1 cells [11,29]. Accordingly, using lipid extraction and subsequent thin layer chromatography, also we detected an increased accumulation of CE as well as lipid droplet formation in HL-1 cells treated with aggLDL for 8 and 18 h, respectively (Figure 1A,B and Appendix A). AggLDL treatment did not significantly affect cell survival (Appendix A). Next, to evaluate whether LRP1 is mediating this lipid accumulation, HL-1 cells were preincubated with GST-RAP, which blocks the binding of all LRP1 ligands, followed by treatment with DiI-labeled aggLDL (aggLDL-DiI) during 8 h and next analyzed by confocal microscopy. As expected, the presence of GST-RAP significantly blocked the uptake of aggLDL (Figure 1C,D). These results support the concept that LRP1 mediates the aggLDL uptake by HL-1 cells producing intracellular CE accumulation.

### 3.2. Aggregated LDL Endocytosis Leads to AggLDL/LRP1 Complex Accumulation in Endosomal Compartments

To evaluate the endocytic pathway that follows LRP1 after binding to aggLDL, HL-1 cells were incubated with aggLDL-DiI for 8 h at 37 °C and then immunostained for detection of LRP1 at different intracellular compartments. Confocal microscopy experiments showed that LRP1 and aggLDL-DiI colocalized in vesicular structures in a high proportion (−50%) (Figure 2A). The co-localization analysis of the different subcellular compartments indicated that main of the LRP1 co-localization with aggLDL occurred in endocytic early endosome [EEA1^+^] (Figure 2B,C) while a weak co-localization occurred in [Rab11^+^] and [Rab4^+^] endocytic recycling as well as trans-Golgi [TGN46^+^] compartments (Figure 2C). These data indicate that aggLDL/LRP1 complex is internalized and accumulated in endocytic compartments in HL-1 cells.

It has been shown that aggLDL prolongs the half-life of LRP1 by inhibiting the proteasomal LRP1 degradation, which in turn favors the uptake and intracellular CE accumulation in VSMC [11]. Therefore, we evaluated the effect of HL-1 exposure to aggLDL for 8 h on LRP1 mRNA and protein levels through quantitative RT-PCR and Western blot assays, respectively. Figure 2D,E shows that aggLDL did not alter mRNA or protein levels of LRP1 at this end point. Therefore, these results demonstrate that aggLDL does not modify the LRP1 protein levels in HL-1 cells.

### 3.3. Aggregated LDL Reduces the Molecular Association Between LRP1 and IR

It has been demonstrated that LRP1 mediates the insulin-induced IR phosphorylation in mouse brain [21]. In addition, hepatic LRP1 deficiency impaired the insulin response of mice fed with high-fat diet [22]. Considering that aggLDL/LRP1 complex is accumulated in endocytic compartments, the cellular functionality of this receptor may be affected. Therefore, we have focused to investigate whether aggLDL may alter the molecular association between LRP1 and IR. In this way, HL-1 cells were incubated with aggLDL for 8 h and then stimulated with insulin for 30 min. Next, immunoprecipitation procedures were carried out using an anti-IR antibody followed by Western blot assays for LRP1 (Figure 3A). The quantitative analysis (Figure 3B) of bands showed that under basal conditions (without insulin) a molecular association is produced between LRP1 and IR (lane 2) that was reduced by aggLDL (lane 3). Insulin increased the LRP1/IR association (lane 5) that was also impaired in the presence of aggLDL (lane 4). Similar results were obtained when immunoprecipitation assays were carried out with anti-LRP1 antibody followed by Western blot assays for IR (Appendix A).

To evaluate whether the inhibitory effect of aggLDL on the molecular association between IR and LRP1 involves the ligand-receptor interaction, immunoprecipitation assays were carried out in cells preincubated with GST-RAP for 30 min. Under basal conditions (without insulin), GST-RAP did not affect the molecular association between LRP1 and IR (lanes 2 and 6) but effectively blocked the aggLDL effect on LRP1/IR association (lanes 3 and 8) (Figure 3C,D). In the presence of insulin, GST-RAP restored the molecular association between LRP1 and IR in cells treated with aggLDL (lanes 4, 5, and 7) (Figure 3C,D). Thus, these results indicate that the molecular association between LRP1 and IR in HL-1 cells is reduced by aggLDL.

### 3.4. Aggregated LDL Impairs Insulin-Induced Intracellular Signaling Activation

Our group has previously demonstrated that insulin promotes the LRP1 translocation to the PM by activation of the insulin-induced intracellular signaling pathway [23,30]. Here, we investigated whether aggLDL may affect both LRP1 and IR levels on the cell surface. For this purpose, HL-1 cells were treated with aggLDL for 8 h, stimulated or not with insulin for 30 min and the levels of both receptors in the PM were analyzed by biotinylation assay of surface proteins followed by Western blot. Figure 4A,B show that insulin as well as aggLDL significantly increased IR and LRP1 protein levels in the PM respect to control. In cells treated with aggLDL and then stimulated with insulin, the levels of LRP1 protein on the PM were increased over the control while those of IR remained unaltered.

Based on these results, we study whether aggLDL affects insulin intracellular signaling activation in HL-1 cells. For this, cells were treated with aggLDL for 8 h, then stimulated with insulin for different times, and the activation of the insulin-induced intracellular signaling was evaluated by Western blot assays for phosphorylated IR (p-IR), Akt (p-Akt) and AS160 (p-AS160). Figure 4C,D shows that aggLDL significantly counteracted the time-dependent insulin-induced phosphorylation of IR, Akt and AS160. To evaluate whether the inhibitory effect of aggLDL on the insulin-induced intracellular signaling involves LRP1, HL-1 cells were incubated with GST-RAP previous to aggLDL and insulin treatments. Figure 4E,F shows that GST-RAP prevented the inhibitory effect of aggLDL on insulin-activated intracellular signaling as evidenced by increased p-Akt levels. These results indicate that aggLDL via LRP1 impairs the insulin intracellular signaling activation in HL-1 cells.

### 3.5. Aggregated LDL Affects the GLUT4 Traffic to the Cell Surface and Glucose Uptake

The insulin-induced intracellular signaling is a critical event for the GLUT4 traffic to the PM, which promotes the uptake of extracellular glucose [19]. HL-1 cells also express GLUT1, which internalizes glucose in an insulin-independent manner [31]. Herein, we found that the GLUT4/GLUT1 ratio is not modified in the presence of aggLDL and insulin in HL-1 cells (Appendix A). Based on this, we evaluated whether aggLDL may affect the GLUT4 traffic to the cell surface in HL-1 cells stimulated with insulin. In this way, cells were treated with aggLDL for 8 h and stimulated with insulin for 30 min and the GLUT4 protein level on the PM was measured by biotinylation assay of surface proteins. Figure 5A,B shows that insulin increased GLUT4 in the PM, whereas aggLDL alone did not produce significant changes on the GLUT4 in the cell surface. Finally, the insulin-induced GLUT4 traffic to the PM was impaired by aggLDL treatment.

Taking into account that aggLDL affects the insulin-induced GLUT4 traffic to the PM, we evaluated the glucose uptake by HL-1 cells. Thus, cells were treated with aggLDL and insulin as above in the presence of 2-NBDG, a glucose fluorescent analogue, for 30 min. Figure 5C,D shows that insulin stimulation significantly increased the 2-NBDG uptake, while aggLDL treatment alone did not alter glucose uptake. However, cells treated with aggLDL evidenced a significant decrease in 2-NBDG uptake in response to insulin. Finally, to explore the potential role of LRP1 in this effect of aggLDL, HL-1 cells were preincubated with GST-RAP for 30 min before cells were exposed to aggLDL, insulin and 2-NBDG treatments. Figure 5E,F shows that the GST-RAP incubation counteracted the impairment effect of aggLDL on the insulin response. These results indicate that aggLDL/LRP1 complex endocytosis affects insulin-induced GLUT4 traffic to the PM and, therefore, the glucose uptake by HL-1 cells.

## 4. Discussion

Frequently, insulin resistance coexists with an imbalance in lipoprotein metabolism which has a strong impact on cardiac metabolism [3]. Previous studies from our group demonstrated LRP1 mediated the CE accumulation in HL-1 cells as well as in neonatal rat ventricular cardiomyocytes (NRVC) [10]. Although HL-1 is a derived cell line from mouse atrial cardiomyocytes and presents certain limitations as an experimental model, numerous research groups have been based on the use of the HL-1 cell line for the study of normal cardiomyocyte function and also in several pathologies [32,33,34]. In the present study, we demonstrated that LRP1 mediates aggLDL binding and endocytosis in this cell line. The aggLDL/LRP1 complex is accumulated in early endosomes [EEA1^+^] but not in other recycling vesicles such as TGN [TGN46^+^] or recycling endocytic compartments [Rab4^+^ and Rab11^+^]. Interestingly, aggLDL-treated HL-1 cells showed a decreased insulin response, which was evidenced by: (i) reduced molecular association between LRP1 and IR; (ii) decreased insulin-induced intracellular signaling (IR, Akt, and AS160 phosphorylation); (iii) impaired GLUT4 traffic to the PM and (iv) reduced extracellular glucose uptake.

In the present work, we show that aggLDL increases intracellular CE accumulation, which is fully blocked by GST-RAP, an intracellular chaperone protein that compete and inhibit the binding of all LRP1 ligands [35]. LRP1 internalizes its ligands by endocytosis mediated by a chlatrin-coated pit process, which is followed by a significant accumulation of the ligand/receptor complex in early endosomes [36]. From this endosomal accumulation, the cargo is routed to a degradation pathway via late endosomes and lysosomes after a few minutes, whereas LRP1 is intracellularly stored in LRP1 storage vesicles (LSVs) and recycling compartments, and subsequently returned to the PM though secretory and exocytic pathways [23,37]. Here, we demonstrate that both aggLDL and LRP1 were mainly localized in endocytic endosomes, which suggest that aggLDL modifies LRP1 intracellular traffic and, therefore, may potentially influence LRP1 functions.

It has been demonstrated that LRP1 is located mainly in clathrin-rich membrane domains, whereas IR is localized in lipid rafts/caveolae domains where the receptor signaling pathway downstream is activated by insulin [38,39,40]. In this way, LRP1 seems to be transiently translocated in lipid rafts/caveolae domains under the stimulating effect of insulin [41]. Our team has previously shown that LRP1 membrane compartmentalization is altered in human myocardial samples from dilated cardiomyopathy patients [42]. In the present study, we show that aggLDL counteracted insulin-induced IR sorting to the PM and reduced the molecular association of LRP1 with IR. Together, our results would suggest that aggLDL mainly alters the LRP1 endocytic processing and its insulin-dependent distribution from clathrin domain to lipid rafts/caveolae domain. Likely, further studies are needed to evaluate if impaired LRP1 and IR membrane distribution is caused by increased CE accumulation in membrane rich cholesterol domains.

LRP1 cooperates as a scaffold protein with IR for its adequate phosphorylation by insulin action in neurons and hepatocytes, although the molecular mechanism is not clear yet and requires further investigation [21,22]. The alteration of the LRP1 expression in these cell types results in a defective signaling of the insulin [22,43]. In the present study, we demonstrated that aggLDL counteracts insulin-induced intracellular signaling (IR, Akt, and AS160 phosphorylation), which was restored by the presence of GST-RAP. In turn, this aggLDL impairment effect of insulin signaling had deleterious consequences on insulin-induced GLUT4 traffic to the PM. Our findings corroborated that aggLDL affect the exocytosis of this glucose transporter towards the cell surface, probably due to the alteration in the insulin-induced intracellular signaling. As consequence of the aggLDL inhibitory effect on LRP1-mediated GLUT4 traffic, 2-NBDG uptake was reduced in aggLDL-loaded HL-1 cells. Further studies will be required to know whether aggLDL also affects other insulin actions on lipid and protein metabolism as well as on cellular energy [44,45]. In this last aspect, it would be interesting to investigate the aggLDL effect by stimulation times longer than 8 h, not only to assess the possible damage caused by CE but also by the disruption of the insulin response on cell viability.

## 5. Conclusions

Figure 6 shows the conclusive scheme of our results in which it is represented that the LRP1-mediated aggregated LDL uptake causes intracellular CE accumulation and contributes to altering the insulin response through impairment of insulin signaling, GLUT4 traffic to the PM, and glucose uptake by HL-1 cells.

## Figures and Tables

**Figure 1 cells-09-00182-f001:**
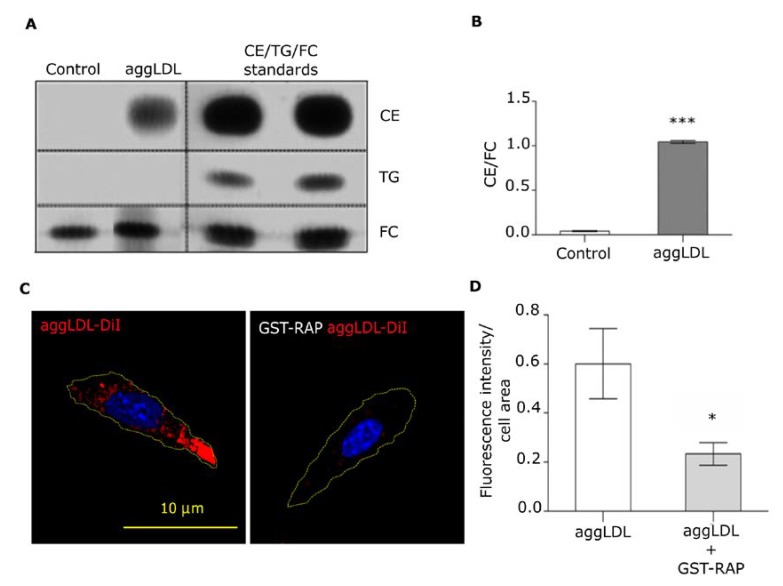
Intracellular lipid content and aggLDL uptake in HL-1 cells. (**A**) Thin layer chromatography assay of lipid extracts from HL-1 cells incubated with 100 µg/mL aggLDL for 8 h. Bands correspond to intracellular cholesteryl ester (CE), triglycerides (TG), and free cholesterol (FC). The CE/TG/FC bands from the standards analyzed by duplicate are shown in the right panel. (**B**) Bar graph represents the mean ± SEM of the optical densities of the bands corresponding to CE relative to the FC content for each condition. The statistical comparison of means was carried out through Student’s *t*-test for independent samples. (***) indicates significant differences (*p* < 0.0001) vs. control. Three independent experiments were performed. (**C**) Confocal microscopy of HL-1 cells preincubated or not with 400 nM GST-RAP and next treated with 100 µg/mL aggLDL-DiI for 8 h. The fluorescent emission of aggLDL-DiI is observed red and yellow dotted line represents the cell shape. The cell nuclei were labeled with Hoechst. The scale bar corresponds to 10 µm. (**D**) Bar graph represents the Mean ± SEM of the Dil fluorescence intensity per cell area. (*) indicates significant differences (*p* < 0.05) between aggLDL treatment with or without GST-RAP. Three independent experiments were performed by duplicate and 20 cells were analyzed per experiment.

**Figure 2 cells-09-00182-f002:**
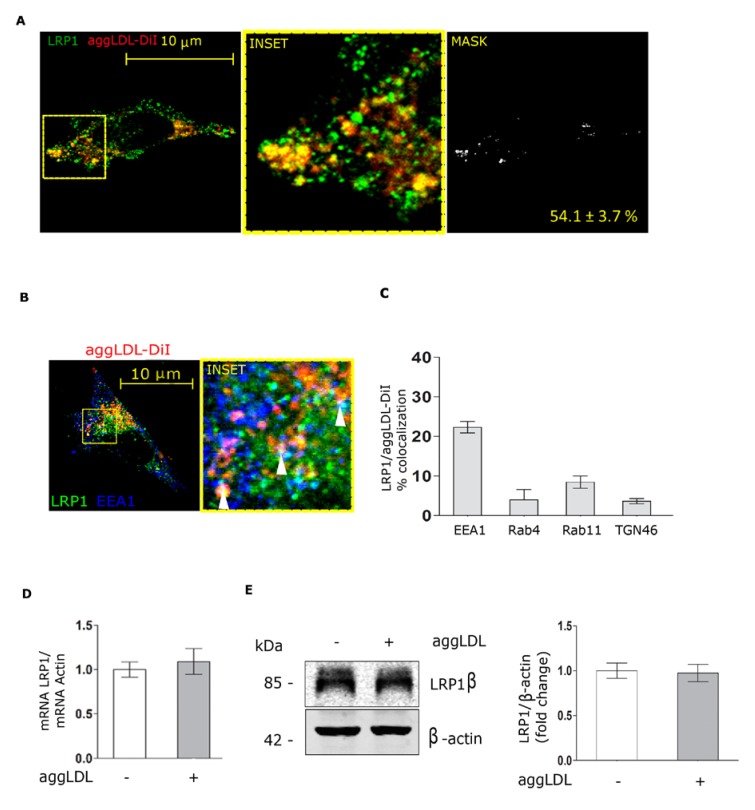
Intracellular distribution and expression level of LRP1 in HL-1 cells. (**A**) Confocal microscopy of HL-1 cells incubated in the absence or presence of 100 µg/mL aggLDL-Dil for 8 h. LRP1 in green and aggLDL-DiI in red are shown. Inset: 4× magnification of framed region in the dotted lines. The MASK represents the binary image where co-localization pixels between LRP1 and aggLDL are shown in white. The corresponding quantitative analysis is shown as the mean ± SD of the percentages of co-localization. The images are representative of 30 cells per condition, coming from three independent experiments. The scale bar corresponds to 10 μm. (**B**) Confocal microscopy in HL-1 cells incubated with 100 µg/mL aggLDL-DiI for 8 h showing the triple co-localization analysis between LRP1 and aggLDL-Dil together with early endosome marker [EEA1^+^]. The inset represents the 4× magnification of the framed regions in the dotted lines. The white arrowheads indicate vesicles containing triple co-localization sectors. The images are representative of 30 cells per condition, coming from three independent experiments. The scale bar corresponds to 10 μm. (**C**) The bar graph represents the mean ± SEM of the co-localization percentages of the triple co-localization between LRP1, aggLDL-Dil and different intracellular compartment markers (EEA1, Rab4, Rab11, and TGN46), obtained from the quantification performed in 30 cells per condition. (**D**) Quantitative RT-PCR assay to evaluate LRP1 mRNA expression in HL-1 cells exposed to 100 μg/mL aggLDL for 8 h. Bar graph showing the Mean ± SEM of mRNA levels. The RT-PCR products of Actin transcripts were used as loading controls. (**E**) Western blot assay for the immunodetection of the LRP1 β-subunit in HL-1 cells treated with 100 μg/mL aggLDL for 8 h. The immunodetection of β-actin was used as a protein loading control. The bar graph represents the relative intensity of LRP1 band normalized to β-actin and reported as a fold change against controls from three different experiments performed in triplicate.

**Figure 3 cells-09-00182-f003:**
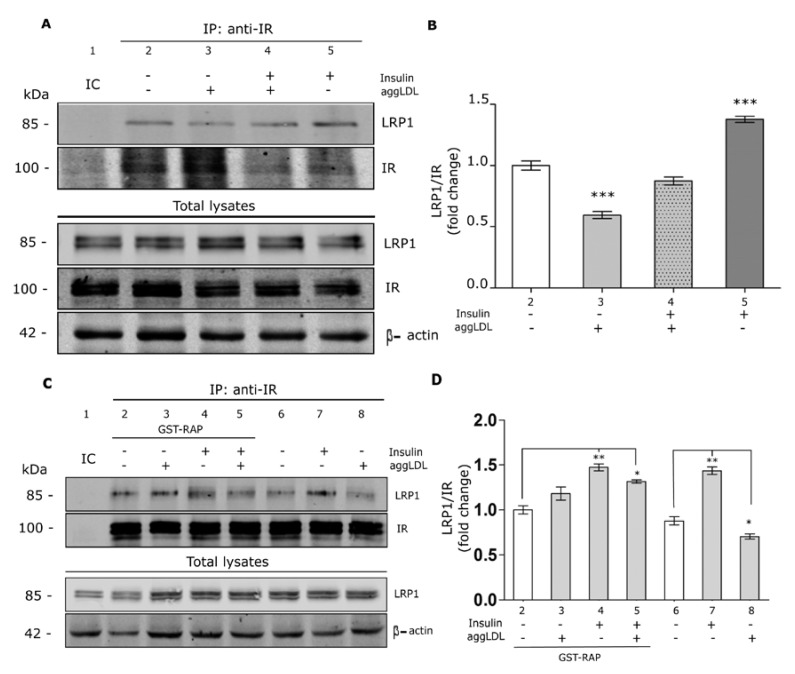
Molecular association between IR and LRP1 in aggLDL-treated HL-1 cells. (**A**) Immunoprecipitation assay for IR. Cell lysate was immunoprecipitated with an anti-IR antibody and then subjected to SDS-PAGE and followed by Western blot assays for LRP1 and total IR. Cells were incubated or not with 100 µg/mL aggLDL for 8 h and then stimulated or not with 100 nM insulin for 30 min. Lane 1, immunoglobulin isotype control (IC). The detection of LRP1, IR and β-actin in total cell lysates is shown in the lower panel of the Western blot. (**B**) The bar graph represents the densitometric quantification of the bands obtained in the Western blot. Values were expressed as the mean ± SEM of the relative intensity of LRP1 bands with respect to IR and shown as a fold change against control (white bar). (***) indicate significant differences (*p* < 0.0001) vs. control. Three independent experiments were performed in duplicate. (**C**) Immunoprecipitation assays for IR followed by Western blot for LRP1 β-subunit. Cells were or not preincubated with 400 nM GST-RAP for 30 min, then incubated or not with 100 µg/mL aggLDL for 8 h in the presence of 400 nM GST-RAP and subsequently stimulated or not with 100 nM insulin for 30 min. Lane 1, immunoglobulin isotype control (IC). The detection of LRP1 and β-actin in the total cell lysates is shown in the lower panel of the Western blot. (**D**) The bar graph represents the densitometric quantification of the bands obtained in the Western blot. The values were expressed as Mean ± SEM of the relative intensity of LRP1 bands with respect to IR, and reported as fold change against the corresponding control (white bar). (*) (*p* < 0.05) and (**) (*p* < 0.01) indicate significant difference regarding to controls. Three independent experiments were performed by duplicate.

**Figure 4 cells-09-00182-f004:**
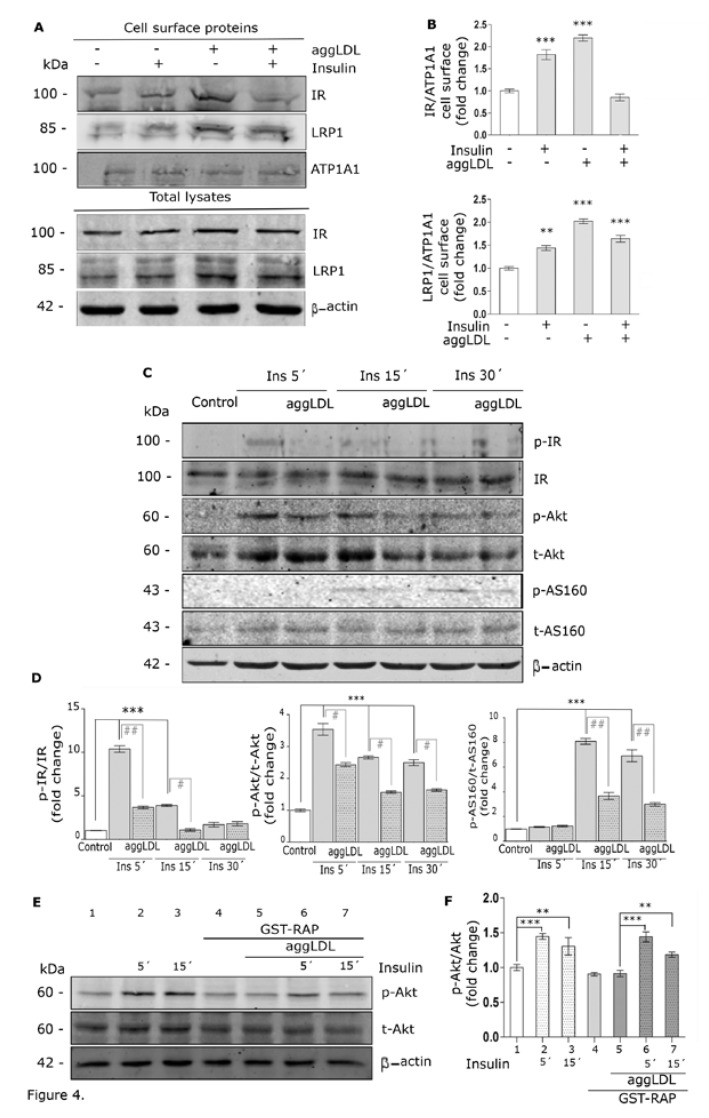
LRP1 and IR protein levels in the PM and insulin-induced intracellular signaling activation in HL-1 cells. (**A**) Biotinylation assay of surface LRP1 and IR proteins. Cells were incubated with 100 µg/mL aggLDL for 8 h and subsequent stimulated with 100 nM insulin for 30 min. Biotinylated proteins were isolated from protein extracts through streptavidin-conjugated bead pull-down. The lower panel of the Western blot shows the detection of LRP1 and IR in total lysates, where 20% of the total protein content was loaded for the incubation with beads. Biotin-ATP1A1 and β-actin were used as protein loading controls of PM and total protein extracts, respectively. (**B**) Bar graphs show the mean ± SEM of the relative optical densities (O.D.) (referred to biotin-ATP1A1 as loading control of PM proteins) of the bands corresponding to LRP1 and IR of cell surface, expressed as fold change against the control (white bar). (***) (*p* < 0.001) and (**) (*p* < 0.01) indicate significant difference respect to control. Three independent experiments were performed by triplicate. (**C**) The Western blot assay of phosphorylated IR (p-IR), Akt (p-Akt) and AS160 (p-AS160) relative to each total protein in HL-1 cells treated with aggLDL as above and stimulated or not with 100 nM insulin for 5 min to 30 min. The immunodetection of β-actin was used as a protein loading control. (**D**) Bar graphs represent the densitometric quantification of the bands obtained in the Western blot results. Respective values were expressed as the mean ± SEM of the relative intensity of p-IR/IR, p-Akt/t-Akt or p-AS160/t-AS160 represented as fold change against the controls (white bar). (***) (*p* < 0.001) indicates significant increase regarding to control. (#) (*p* < 0.05) and (##) (*p* < 0.01) indicate significant difference between the indicated conditions. Three independent experiments were performed by duplicate. (**E**) Western blot assay for phosphorylated Akt (p-Akt) and total Akt (t-Akt) in HL-1 cells preincubated or not with 400 nM GST-RAP for 30 min, treated or not with 100 µg/mL aggLDL for 8 h and stimulated or not with 100 nM insulin for 5 min and 15 min. The immunodetection of β-actin was used as a protein loading control. (**F**) Bar graph represents the densitometric quantification of the bands. Values were expressed as the mean ± SEM of the relative intensity of p-Akt/t-Akt, reported as fold of change respect to the corresponding controls (lanes 1 and 5). (**) (*p* < 0.01) and (***) (*p* < 0.0001) indicate significant difference respect to cells treated or not with aggLDL and with or without GST-RAP. Three independent experiments were performed by duplicate.

**Figure 5 cells-09-00182-f005:**
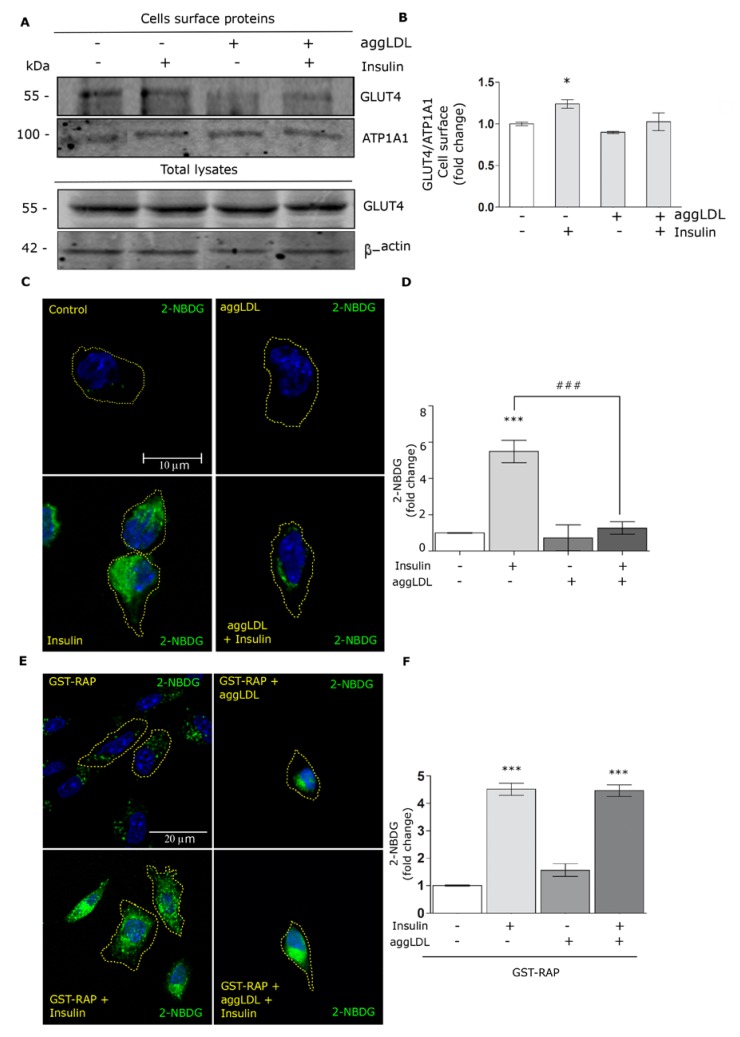
Level of GLUT4 protein in the plasma membrane and 2- NBDG uptake by HL-1 cells treated with aggLDL and insulin. (**A**) Biotinylation assay of surface proteins for the detection of biotinylated GLUT4, from protein extracts after an incubation assay with 100 µg/mL aggLDL for 8 h and subsequent stimulation with 100 nM insulin for 30 min. Next, biotinylated proteins on the cell surface were subsequently isolated by incubation with streptavidin-conjugated beads. The GLUT4 detection in the total cell lysates is shown in the lower panel of the Western blot. ATP1A1 and β-actin were used as protein loading controls, respectively. (**B**) Bar graph represents the mean ± SEM of the relative optical densities (O.D.) (referred to biotin-ATP1A1 as loading control of PM proteins) of the bands corresponding to GLUT4 at the cell surface for the different conditions, expressed as the fold change against controls (white bar). (*) (*p* < 0.05) indicates significant difference against controls. Three independent experiments were performed by triplicate. (**C**) Confocal microscopy of HL-1 cells incubated or not with 100 µg/mL aggLDL for 8 h and stimulated or not with 100 nM insulin for 30 min, together with 80 μM 2-NBDG (green). The images are representative of 20 cells per condition from three independent experiments. The scale bar corresponds to 10 μm. Dotted line represents the cell shape. (**D**) Bar graph represents the mean ± SEM of the fluorescence intensity of 2-NBDG per cell area, expressed as the fold chances against controls (white bar). (***) (*p* < 0.001) indicates the significant differences against controls. (###) (*p* < 0.001) indicates significant difference between the indicated conditions. (**E**) Confocal microscopy of HL-1 cells preincubated with 400 nM GST-RAP and treated or not with 100 µg/mL aggLDL for 8 h. Then, cells were stimulated or not with 100 nM insulin for 30 min, together with 80 μM 2-NBDG (green). Images are representative of 20 cells per condition. The scale bar corresponds to 20 μm. The dotted line represents the cell shape. (**F**) Bar graph represents the mean ± SEM of the fluorescence intensity of 2-NBDG per cell area, expressed as fold change against controls (white bar). (***) (*p* < 0.0001) indicates significant difference respect to control. Three independent experiments in duplicate were performed.

**Figure 6 cells-09-00182-f006:**
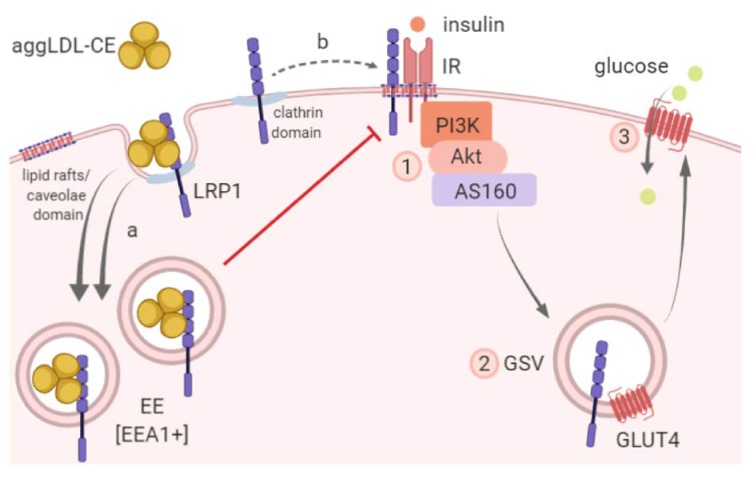
LRP1-mediated aggregated LDL uptake and impaired insulin response in HL-1 cells. LRP1 mediates the binding and endocytosis of aggLDL (indicated as (**a**) causing intracellular cholesteryl ester (CE) accumulation. In the present work we demonstrated that the LRP1/aggLDL complex is located mainly in early endosomes [EEA1+], which could have consequences in LRP1 insulin-dependent distribution from clathrin domain to lipid rafts/caveolae domain (indicated as **b**). In addition, we showed that the aggLDL accumulation also modified the molecular association between LRP1 and IR, which could contribute to an impaired insulin response characterized by (1) altered insulin-induced intracellular signaling activation, (2) reduced GLUT4 traffic to the PM, and (3) decreased glucose uptake. EE, early endosomes; GLUT4, glucose transporter 4; GSV, GLUT4 storage vesicles.

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
