# Peer review of "LRP1-Mediated AggLDL Endocytosis Promotes Cholesteryl Ester Accumulation and Impairs Insulin Response in HL-1 Cells"

_cells, 2020, doi:10.3390/cells9010182_

Round 1

Reviewer 1 Report

Dato et al describe a comprehensive study using different methods in a murine cardiac cell line (HL-1) the relationship between lipoprotein receptor-related protein-1 (LRP1), receptor-mediated endocytosis and insulin resistance. 

With a series of experiments and technical approaches, they concluded that LRP1-aggregated LDL endocytosis is associated with insulin resistance as measured by Akt phosphorylation and glut-4 translocation.

I have some few comments:

in material and methods: although the antibody catalog number is mentioned, the phosphoprotein antibodies should mention the aminoacids.

Because HL-1 is a mouse cell line, how the authors control the staining with the mouse primary antibodies? Ex: anti-APT1A1 

line 147 PFA should be spelled out

Figure 2E, 4A and 5B: the b-actin appears to be from a different membrane of Lpr1B.

Although I would highly recommend figure 10, the actual figure does not crearly describes what the main results are and the legends are incomplete. Ex: spell out CE, LRP1, etc....

These results, obtained from a mouse cell line of putative HL-1 cells is far from suggestioin clinical implications and therapeutic connotations. Further studies pre-clinical in vivo and clinical are needed before a suggestion of this sort can be made. Please keep your conclusions to the data obtained from a cell line of questionable origin.

Author Response

Reviewer #1

Dato et al describe a comprehensive study using different methods in a murine cardiac cell line (HL-1) the relationship between lipoprotein receptor-related protein-1 (LRP1), receptor-mediated endocytosis and insulin resistance. 

With a series of experiments and technical approaches, they concluded that LRP1-aggregated LDL endocytosis is associated with insulin resistance as measured by Akt phosphorylation and glut-4 translocation.

I have some few comments:

1) in material and methods: although the antibody catalog number is mentioned, the phosphoprotein antibodies should mention the aminoacids.

Answer: Phosphorylated residues of different proteins have been included in Material and Methods (section 2.1) in the marked revised version (lines 92, 94, and 95).

2) Because HL-1 is a mouse cell line, how the authors control the staining with the mouse primary antibodies? Ex: anti-APT1A1

Answer: These primary antibodies were used to detect mouse proteins by Western blot assays. Considering that HL-1 cells do not produce immunoglobulins (IgG), is not possible immune-crossreaction with anti-mouse secondary antibodies. For IF-microscopy assays the blocking steps of unspecific sites are described in Materials and Methods (section 2.6). In general, all primary antibodies originating in mouse having reactivity with mouse antigens as indicated by their data sheet provided by respective commercial companies.

3) line 147 PFA should be spelled out

Answer: It has already been modified in the revised version in Materials and Methods (section 2.6; marked revised version: line 149).

4) Figure 2E, 4A and 5B: the b-actin appears to be from a different membrane of Lpr1B.

Answer: Both bands come from the same membranes in each of the indicated images, but have been accidentally trimmed differently in the assembly of the figures. This has already been modified in the revised figures (Fig. 2E, 4A, and 5B)

5) Although I would highly recommend figure 10, the actual figure does not crearly describes what the main results are and the legends are incomplete. Ex: spell out CE, LRP1, etc....

Answer: We think that the reviewer comment is referred to Fig. 6 instead Fig.10. In this way, and considering the recommendation of the reviewer, we have modified the legend of figure 6 as follow:

Figure 6. LRP1-mediated aggregated LDL uptake and impaired insulin response in HL-1 cells. LRP1 mediates the binding and endocytosis of aggLDL (indicated as a) causing intracellular cholesteryl ester (CE) accumulation. In the present work we demonstrated that the LRP1/aggLDL complex is located mainly in early endosomes [EEA1+], which could have consequences in LRP1 insulin-dependent distribution from clathrin domain to lipid rafts/caveolae domain (indicated as b). In addition, we showed that the aggLDL accumulation also modified the molecular association between LRP1 and IR, which could contribute to an impaired insulin response characterized by 1) altered insulin-induced intracellular signaling activation, 2) reduced GLUT4 traffic to the PM and 3) decreased glucose uptake. EE, early endosomes; GLUT4, glucose transporter 4; GSV, GLUT4 storage vesicles.

6) These results, obtained from a mouse cell line of putative HL-1 cells is far from suggestion in clinical implications and therapeutic connotations. Further studies pre-clinical in vivo and clinical are needed before a suggestion of this sort can be made. Please keep your conclusions to the data obtained from a cell line of questionable origin.

Answer: We agreed with the limitations of the experimental model. Thus, our conclusions were limited to results obtained with HL-1 cells, which have been mainly modified in the Discussion and Conclusion Sections of the revised version.

Reviewer 2 Report

Dato, et al., have studied the effects of LRP1 mediated endocyotsis aggregated LDL (aggLDL) on cholesterol ester (CE) accumulation and insulin response. Evidence is presented for LRP1 mediation of aggLDL endocytosis, CE accumulation and imparied insulin response in the presence of aggLDL.

Concerns:

The results presented are very interesting. Did the authors investiagate hypoxic conditions?

Although GLUT4 is typically the dominant isoform of glucose transporters in the myocardium, GLUT1 expression is known to increase in certain pathological states when GLUT4 is repressed. Did the authors look at GLUT1 expression in response to aggLDL treatment in the presence and absence of insulin?

Figure S1, panel B shows no difference in cell viability upon exposure to aggregated LDL. Did the authors test TGF-β levels in cells treated with aggLDL? It has been shown in the literature that TGF- β levels are elevated in myocardial infarction and that LRP1 is involved in the TGF-β signalling.

Related to the previous question – although the cell viability was not affected, were there differences in cellular energetics (e.g. oxygen consumption, mitochondrial function, etc.)?

It is very difficult to read panels A and B of Figure 2. The confocal microscopy images should be made bigger to make it easier to interpret the images.

In panel A of Figures 3 and 4, why are there two bands present for LRP1? Is that a contamination? How was the quantification of the bands handled?

Panel C of Figure 4 is concerning. The western blot of phosphorylated AS160 is not clear. It appears that it would be rather difficult to quantify the bands from such a western blot as the authors have done. A better western blot needs to be presented. I have similar concerns for p-Akt as well.

Minor issues:

Line 88: Units for the concentration of nor-epinephrine is incorrect

Several typographical errors and missing words are scattered throughout the manuscript.

Author Response

Reviewer #2

Dato, et al., have studied the effects of LRP1 mediated endocyotsis aggregated LDL (aggLDL) on cholesterol ester (CE) accumulation and insulin response. Evidence is presented for LRP1 mediation of aggLDL endocytosis, CE accumulation and imparied insulin response in the presence of aggLDL.

Concerns:

1) The results presented are very interesting. Did the authors investiagate hypoxic conditions?

Answer: In agreement with the reviewer question, hypoxia is a condition where LRP1 is regulated in different types of cells. In particular with our experimental model, previous studies have demonstrated that hypoxia can up-regulate the LRP1 expression in HL-1 cells and neonatal rat ventricular cardiomyocytes (NRVC), inducing a significant CE accumulation by the internalization of aggregated LDL (aggLDL) (Cal et al. Cardiovasc. Res 2012), referenced in the original version as #10. For this reason, it is possible that hypoxia may also affect the insulin response in HL-1 cells. In this way, in our lab we have carried out preliminary experiments under hypoxic conditions in HL-1 cells that surely will be part of another work.

2) Although GLUT4 is typically the dominant isoform of glucose transporters in the myocardium, GLUT1 expression is known to increase in certain pathological states when GLUT4 is repressed. Did the authors look at GLUT1 expression in response to aggLDL treatment in the presence and absence of insulin?

Answer: We agree with the reviewer comment about GLUT1 in cardiomyocytes. However, our interest in the present study was evaluate if the reduced molecular association between LRP1 and IR as well as the impaired IR signaling induced by aggLDL in HL-1 cells could affect certain molecular targets of insulin response, such as GLUT4 and glucose uptake. Considering our findings with GLUT4, in the future surely may be interesting to investigate another glucose transporter such as GLUT1 in cardiomyocytes treated with aggLDL.

3) Figure S1, panel B shows no difference in cell viability upon exposure to aggregated LDL. Did the authors test TGF-β levels in cells treated with aggLDL? It has been shown in the literature that TGF- β levels are elevated in myocardial infarction and that LRP1 is involved in the TGF-β signalling.

Answer: In our laboratory we haven’t test TGF-ß levels in HL-1 cells treated with aggLDL. But, similar to hypoxic conditions mentioned above (answer #1) further studies may be addressed to investigate whether aggLDL may induce the expression of TGF-ß. In agreement with the reviewer comment it would be very interesting for future works.

4) Related to the previous question – although the cell viability was not affected, were there differences in cellular energetics (e.g. oxygen consumption, mitochondrial function, etc.)?

Answer: We haven’t evaluated it yet but it would be interesting to estimate the mitochondrial function in cells that accumulate CE since in other models it has been shown that the uptake of cholesterol generates alteration of the mitochondrial function (Solsona-Vilarrasa et. al, Redox Biol. 2019). To consider the reviewer’s comment in discussion we included a phrase that contemplate this aspect to be investigated in the future, as follow:

Discussion; Line 526-527: “Further studies will be required to know whether aggLDL also affects other insulin actions on lipid and protein metabolism as well as on cellular energy [43, 44]”; in marked revised version; Ref #44: Solsona-Vilarrasa et al. Redox Biol. 2019.

5) It is very difficult to read panels A and B of Figure 2. The confocal microscopy images should be made bigger to make it easier to interpret the images.

Answer: It has already been modified in the final revised figures.

6) In panel A of Figures 3 and 4, why are there two bands present for LRP1? Is that a contamination? How was the quantification of the bands handled?

Answer: It is known that subunit-ß of LRP1 can be synthetized as two bands of 85 and 90 kDa in which by Western blot appear as a couple band (Aschom et al. J Cell Biol 1989). Thus, for quantification both bands were considered for densitometric analysis.

7) Panel C of Figure 4 is concerning. The western blot of phosphorylated AS160 is not clear. It appears that it would be rather difficult to quantify the bands from such a western blot as the authors have done. A better western blot needs to be presented. I have similar concerns for p-Akt as well.

Answer: Considering the reviewer’s comment the panel C of Figure 4 was modified to obtain a best visualization of bands.

Minor issues:

8) Line 88: Units for the concentration of nor-epinephrine is incorrect

Answer: This aspect has been corrected.

9) Several typographical errors and missing words are scattered throughout the manuscript.

Answer: The grammar of the text has been revised

Reviewer 3 Report

In the current study the authors add further progress to their recent findings about Lrp-1 and impaired insulin/Glut-4 signaling in HL-1 cells. The authors describe in a set of experiments that aggLDL can attenuate the described coupling between Lrp1 and the insulin receptor and speculate about the relevance in patients with diabetes and metabolic syndrome.

Major comment: All experiments with HL-1 cells are carefully performed and nicely presented. The main point I have with this manuscript is that the authors completely ignore the fact that they are working with a cell line derived from mice atrial cells years ago. Although I agree that this cell line is sufficient to investigate the relationship between Lrp1 and insulin receptor and signaling, I completely disagree with the authors conclusions that HL-1 are representative for cardiomyocytes. The citation used in this manuscript (ref. 10) shows in contrast to the statement in the introduction that normal, normoxic neonatal rat cardiomyocytes express MORE LDL receptor than Lrp-1. Moreover a recent study using terminal differentiated adult rat cardiomyocytes (Basic Res. Cardiol. 112, 63) nicely shows that these cells express mainly oxLDL receptors and that in agreement with ref. 10 rat cardiomyocytes express more LDL receptors than Lrp-1. In other words, the conclusion that we can learn from these HL-1 cells anything about cardiac physiology under conditions of disturbed CE metabolism are misleading. At present, it may be a different between species (mouse-rat), location (atrium-ventricle) or artificial changes in metabolism as these cells are tumor cells. Therefore, the authors cannot discuss their project in this context.

Author Response

Reviewer #3:

In the current study the authors add further progress to their recent findings about Lrp-1 and impaired insulin/Glut-4 signaling in HL-1 cells. The authors describe in a set of experiments that aggLDL can attenuate the described coupling between Lrp1 and the insulin receptor and speculate about the relevance in patients with diabetes and metabolic syndrome.

Major comment: All experiments with HL-1 cells are carefully performed and nicely presented.

1) The main point I have with this manuscript is that the authors completely ignore the fact that they are working with a cell line derived from mice atrial cells years ago. Although I agree that this cell line is sufficient to investigate the relationship between Lrp1 and insulin receptor and signaling, I completely disagree with the authors conclusions that HL-1 are representative for cardiomyocytes. The citation used in this manuscript (ref. 10) shows in contrast to the statement in the introduction that normal, normoxic neonatal rat cardiomyocytes express MORE LDL receptor than Lrp-1. Moreover a recent study using terminal differentiated adult rat cardiomyocytes (Basic Res. Cardiol. 112, 63) nicely shows that these cells express mainly oxLDL receptors and that in agreement with ref. 10 rat cardiomyocytes express more LDL receptors than Lrp-1. In other words, the conclusion that we can learn from these HL-1 cells anything about cardiac physiology under conditions of disturbed CE metabolism are misleading. At present, it may be a different between species (mouse-rat), location (atrium-ventricle) or artificial changes in metabolism as these cells are tumor cells. Therefore, the authors cannot discuss their project in this context.

Answer: We completely agree with the Reviewer that HL-1 cells in basal conditions express higher levels of LDL-receptor than LRP1. However, under hypercholesterolemic-like conditions, LDL receptor is downregulated while LRP1 is upregulated in HL-1 cells like in rat cardiomyocytes. In the presence of aggregated LDL, LRP1 has been reported to be up-regulated in different cell types including human coronary vascular smooth muscle cells (Llorente-Cortes et al, Circulation 2002; JMB 2006; Atherosclerosis 2010) and HL-1 cells (Cal R et al, JTM 2012; Samouillan V et al, IJBCB 2014). Unfortunately, we have incorrectly mentioned this concept referred with the expression levels between LRP1 and LDLR in HL-1 cells and NRVC (referenced in the original version as #10). Therefore, this wrong mention has been solved in the marked revised version of manuscript, deleting this paragraph (line 55).

Although we agree with the reviewer regarding the limitations of the experimental model used in our study, numerous research groups have been based on the use of the HL-1 cell line as an experimental model for the study of normal cardiomyocyte function and also in several pathologies. In this way, HL-1 cells have been used to evaluate normal signaling activation of different pathways (McWhinney et. al, Mol Cell Biochem 2000; Kitta K et. al; Biochem J 2001; Seymour EM, et. al; J Surg Res 2003), electrophysiology (Claycomb WC et. al; Proc Natl Acad Sci U S A. 1998; Kupershmidt et. al, FASEB 2003), metabolic function (Huang et. al, Peptides. 2016) and transcriptional regulation (Chahwan C et. al, Mol Cell Biol 2003; Lanson, NA et. al; Nucleic Acids Res 2000.). More recently, this cell line has been used to evaluate pathologic process such as cardiac remodelling, inflammation and fibrosis (Thirugnanam et. al; J Am Heart Assoc. 2019; Zhuang et. al, Eur Rev Med Pharmacol Sci. 2017; Rawal et. al, Cardiovasc Diabetol. 2019), hypoxia/reoxygenation injury (Wang et. al, J Tissue Eng Regen Med. 2019; Li et. al, Cardiovasc Res. 2019), and imbalances in metabolism and electrophysiology (Lee TI et. al, J Cell Mol Med. 2019; Revuelta-López et. al, J Mol Cell Cardiol. 2015; Lugenbiel et. al, Cell Physiol Biochem. 2018).  These antecedents have been included in the marked revised version (Discussion, line 445 to 448). Although we agree that this experimental model has certain limitations, we believe that the in vitro evidence provided by our work is an important indication for future research related to the heart's energy metabolism. Finally, and in agreement with the reviewer, we avoid overstating the conclusions of our results by limiting interpretations to the findings obtained to the cellular level (in our case, HL-1 cells), eluding to mention clinical and therapeutic implications as well as human pathological extrapolations of our results. These modifications are detailed along the text in the Introduction and Discussion sections of the revised version of manuscript.

Round 2

Reviewer 2 Report

Authors have addressed most of my concerns. Unfortunately, concerns about relative expression levels of GLUT-4/GLUT-1 (glucose transporters, in general) in this system and cellular energetics are still present.

While it is true that these aspects may be addressed in a future study, these aspects are of particular relevance to this study as presented. For example, the authors only need to check if the GLUT4 expression is much higher than GLUT1.

Similarly, it is important to understand the effects of aggLDL-CE accumulation on cellular energetics. Cells used in the study were treated with aggLDL for 8 hours. The work (ref #44) cited by authors suggests cholesterol accumulation disrupts OxPhos and respiratory complexes, albeit in the liver. Oxygen consumption, for example, at specific time points during those eight hours can on the health of the cells. If the oxygen consumption decreases over the 8 hour period, it would be an important marker for cellular viability.

Author Response

Authors have addressed most of my concerns. Unfortunately, concerns about relative expression levels of GLUT-4/GLUT-1 (glucose transporters, in general) in this system and cellular energetics are still present.

While it is true that these aspects may be addressed in a future study, these aspects are of particular relevance to this study as presented. For example, the authors only need to check if the GLUT4 expression is much higher than GLUT1.

Answer: This aspect has already been included in the marked version (section 3.5; marked revised version: line 392 and in Figure S1E and F). By Western blot assays we found that both GLUT4 and GLUT1 are expressed in HL-1 cells in which the GLUT4/GLUT1 ratio is not modified in the presence of aggLDL (8h) and insulin (30 min) (Figure S1E and F). Concerns about relative expression levels of GLUT4/GLUT1, and considering similar antibodies affinities and avidities, seem that GLUT4 is higher expressed than GLUT1 in HL-1 cells.  Nevertheless, it has been demonstrated that GLUT1 internalizes different substrates, including glucose, and does it through an independent insulin mechanism (Mueckler M, Thorens B. The SLC2 (GLUT) family of membrane transporters. Mol Aspects Med. 2013, 34(2-3):121-38. doi: 10.1016/j.mam.2012.07.001). Therefore, in the present work we focus on the GLUT4 traffic to PM since this transporter is the only one regulated exclusively by insulin, so this transporter would help us to evaluate the deterioration of the insulin response as its trafficking to PM occurs early after activation of the insulin signaling pathway.

Similarly, it is important to understand the effects of aggLDL-CE accumulation on cellular energetics. Cells used in the study were treated with aggLDL for 8 hours. The work (ref #44) cited by authors suggests cholesterol accumulation disrupts OxPhos and respiratory complexes, albeit in the liver. Oxygen consumption, for example, at specific time points during those eight hours can on the health of the cells. If the oxygen consumption decreases over the 8 hour period, it would be an important marker for cellular viability.

Answer: This aspect has already been included in the marked version (section 4; marked revised version: line 482). In ref #44 they used mice fed a high cholesterol diet for two days and then the liver mitochondria were isolated to evaluate CE accumulation and mitochondrial function. Both, cellular models and cholesterol stimulation times differ widely between ref #44 and our work, so it would not be entirely correct to try to extrapolate the results of ref #44 in our experimental model. In any case, we believe that it would be important to evaluate the effect of aggLDL on energy metabolism by stimuli longer than 8 hours with lipoproteins, not only to assess the possible damage caused by CE but also by the disruption of the insulin response, which has been shown to have an effect on mitochondrial functionality. The main objective of our work was to evaluate early events after the disruption of the insulin response and at least during that period we did not find a significant decrease in cell viability.